# Integral Representations of a Generalized Linear Hermite Functional

Roberto S. Costas-Santos 

Department of Quantitative Methods, Universidad Loyola Andalucía, E-41704 Seville, Spain; rscosa@gmail.com

**Abstract:** In this paper, we find new integral representations for the generalized Hermite linear functional in the real line the complex plane. As an application, new integral representations for the Euler Gamma function are given.

**Keywords:** integral representation; Hermite functions; generalized hermite linear functional; gamma function

## 1. Introduction

The integral representation of special functions provides an alternative way to express these functions in terms of integrals involving other functions. They often involve a weight function and a kernel function related to the specific special function being considered. The weight function appears as a factor in the integral and reflects the orthogonality property of the associated orthogonal polynomials, and the kernel function represents the additional dependence.

The integral representation allows us to express special functions as infinite series or integrals involving some classical orthogonal polynomials. This connection arises from the fact that the orthogonality condition is satisfied by classical orthogonal polynomials, which naturally leads to the appearance of these polynomials in the integral representation of special functions. In this work, we are going to consider the Hermite polynomials.

The hypergeometric functions, which have applications in many areas, including mathematical physics and combinatorics, can be represented in terms of integrals involving other hypergeometric functions and classical orthogonal polynomials like the Jacobi, Hermite, and Laguerre polynomials, which can be expressed as hypergeometric series (see c.f. [1] and [2] (Section 16)).

For a detailed history of the subject of integral representations for hypergeometric series and basic hypergeometric functions (which is a natural extension of the hypergeometric series), see [3] and [4] (Chapter 4).

R. Sfaxi has established in [5], by means of a linear isomorphism, the so-called *intertwining operator* on polynomials, a relationship between the ordinary Hermite polynomials and their analog nonsingular and of Laguerre–Hahn with class zero. Among others, the author has put in value an important linear functional, namely *the generalized Hermite linear functional*, denoted by $\mathscr{G}_H(\tau)$ of index $\tau \in \mathbb{C}$, with $\tau \neq -n$, $n \geq 1$, where their moments are given by

$$\left(\mathscr{G}_H(\tau)\right)_n := \langle \mathscr{G}_H(\tau), x^n \rangle = \begin{cases} \dfrac{(\tau+1)_{2k}}{k! 2^{2k}}, & \text{if } n = 2k, \\ 0, & \text{if } n = 2k+1, \end{cases} \tag{1}$$

where $(a)_n$ is the Pochhammer symbol, defined as

$$(a)_0 := 1, \quad (a)_k := a(a+1)\cdots(a+k-1), \quad a \in \mathbb{C} \setminus \{0\}, \ k = 1, 2, 3, \ldots,$$

thus $\mathscr{G}_H(\tau)$ is symmetric and monic, i.e., $\left(\mathscr{G}_H(\tau)\right)_0 = 1$.

Observe that setting $\tau = 0$ in (1) we recover the Hermite linear functional, i.e., $\mathscr{G}_H \equiv \mathscr{G}_H(0)$, that is well-known by its integral representation

$$\langle \mathscr{G}_H, p \rangle = \frac{1}{\sqrt{\pi}} \int_{-\infty}^{\infty} p(x) e^{-x^2} dx, \quad p \in \mathbb{P}. \tag{2}$$

So we can write

$$\left(\mathscr{G}_H(\tau)\right)_n = \frac{(\tau+1)_n}{(1)_n} \left(\mathscr{G}_H\right)_n, \quad n = 0, 1, \ldots$$

Note that the linear functional $\mathscr{G}_H$ is classical, since it is quasi-definite and satisfies the Pearson equation

$$\mathscr{G}_H' + 2x\mathscr{G}_H = 0. \tag{3}$$

Taking this into account, the following result holds.

**Lemma 1.** *For any $\tau \in \mathbb{C}$, the linear functional $\mathscr{G}_H(\tau)$ fulfills the difference equation*

$$\left(x^2 \mathscr{G}_H(\tau)\right)'' + \left(2x(x^2 - \tau - 2)\mathscr{G}_H(\tau)\right)' + \left(-4x^2 + (\tau+1)(\tau+2)\right)\mathscr{G}_H(\tau) = 0.$$

**Proof.** Let $\tau \in \mathbb{C}$, if we define the linear functional $\mathscr{E}(\tau)$ as

$$\mathscr{E}(\tau) := \left(x^2 \mathscr{G}_H(\tau)\right)'' + \left(2x(x^2 - \tau - 2)\mathscr{G}_H(\tau)\right)' + \left(-4x^2 + (\tau+1)(\tau+2)\right)\mathscr{G}_H(\tau).$$

Then, for $n \geq 0$, one obtains

$$\left(\mathscr{E}(\tau)\right)_n = -2(n+2)\left(\mathscr{G}_H(\tau)\right)_{n+2} + (n+\tau+2)(n+\tau+1)\left(\mathscr{G}_H(\tau)\right)_n. \tag{4}$$

Since $\mathscr{G}_H(\tau)$ is symmetric, then $\left(\mathscr{E}(\tau)\right)_{2k+1} = 0$, for every $k \geq 0$. On the other hand, setting $n = 2k$ in (4) and taking into account (1), we get for $k \geq 0$,

$$\begin{aligned}
\left(\mathscr{E}(\tau)\right)_{2k} &= -4(k+1)\left(\mathscr{G}_H(\tau)\right)_{2k+2} + (2k+\tau+2)(2k+\tau+1)\left(\mathscr{G}_H(\tau)\right)_{2k} \\
&= -\frac{(\tau+1)_{2k+2}}{k!2^{2k}} + \frac{(2k+1+\tau+1)(2k+\tau+1)(\tau+1)_{2k}}{k!2^{2k}} \\
&= 0.
\end{aligned}$$

Therefore, $\left(\mathscr{E}(\tau)\right)_n = 0$ for all $n = 0, 1, \ldots$. Hence, the result holds. $\square$

Our purpose in this work is to provide integral representations for the linear functional $\mathscr{G}_H(\tau)$, either on the real axis, or on the complex plane. More precisely, the problem consists of determining a weight function $G_H(\bullet; \tau)$, such that

$$\langle \mathscr{G}_H(\tau), p \rangle = \int_{\Omega} p(x) G_H(x; \tau) dx, \quad p \in \mathbb{P},$$

where $\Omega$ is an interval in the real line, or a contour in the complex plane.

The paper is organized as follows. In the next section, there are some preliminaries and notations. In Sections 3 and 4, integral representations in the real line and in the complex plane, respectively, are provided. As an application of the previous results, in Section 5, some new integral representations for the Euler Gamma function are given.

## 2. Preliminaries and Notation

Let $\mathbb{P}$ be the vector space of polynomials with complex coefficients and let $\mathbb{P}'$ be its dual space. We denote by $\langle u, f \rangle$ the action of the linear functional $u \in \mathbb{P}'$ on the polynomial $f \in \mathbb{P}$. In particular, we denote by $(u)_n := \langle u, x^n \rangle$, $n \geq 0$, the moments of $u$.

**Definition 1.** *A linear functional $u$ is called symmetric if $(u)_{2n+1} = 0$, for all $n = 0, 1, \ldots$, and it is called monic if $(u)_0 = 1$.*

In fact, for any $\tau \in \mathbb{C}$, the linear functional $\mathscr{G}_H(\tau)$ is symmetric (see (1)) which allows us to suppose the weight function $G_H(\bullet; \tau)$ is even, i.e., it can be written as $G_H(x; \tau) = U(|x|; \tau)$, where $U(\bullet; \tau)$ is a function defined on $(0, \infty)$. In fact, this is a direct consequence of the following result.

**Lemma 2.** *Let $\mathscr{L}$ be a symmetric linear function that has an integral representation. Then, there exists a function $U$ defined on $(0, \infty)$, such that*

$$\langle \mathscr{L}, p \rangle = \int_{-\infty}^{\infty} p(x) U(|x|) dx.$$

**Proof.** From the assumption there exists a function $L$, defined on $(-\infty, \infty)$, such that

$$\langle \mathscr{L}, p \rangle = \int_{-\infty}^{\infty} p(x) L(x) dx.$$

Let us introduce the following two functions, defined on $(0, \infty)$, as follows:

$$U(x) = \frac{L(x) + L(-x)}{2}, \quad V(x) = \begin{cases} \dfrac{L(x) - L(-x)}{2x}, & \text{if } x \neq 0, \\ 0, & \text{if } x = 0. \end{cases}$$

A straightforward calculation gives that $L(x) = U(|x|) + xV(|x|)$, for all $x \in \mathbb{R}$. Moreover, since $x^{2n+1}V(|x|)$ is an odd function we have

$$(\mathscr{L})_{2n} = \int_{-\infty}^{\infty} x^{2n} U(|x|) dx + \int_{-\infty}^{\infty} x^{2n+1} V(|x|) dx = \int_{-\infty}^{\infty} x^{2n} U(|x|) dx.$$

On the other hand, since $\mathscr{L}$ is symmetric and $x^{2n+1}U(|x|)$ is an odd function, we get

$$(\mathscr{L})_{2n+1} = \int_{-\infty}^{\infty} x^{2n+1} U(|x|) dx = 0.$$

Therefore, for any polynomial $p \in \mathbb{P}$,

$$\langle \mathscr{L}, p \rangle = \int_{-\infty}^{\infty} p(x) U(|x|) dx.$$

□

The next result related to hypergeometric functions will be useful later.

**Lemma 3** ([6,7]). *The following formulae hold:*
1. *If $\Re(\alpha) > 0$ and $\Re(s) > 0$, then*

$$\int_0^{\infty} t^{\alpha-1} {}_1F_1(a_1; b_1; t) e^{-st} dt = \frac{\Gamma(\alpha)}{s^\alpha} {}_2F_1(a_1, \alpha; b_1; 1/s). \tag{5}$$

2. If $\Re(c - a - b) > 0$, then

$$_2F_1(a,b;c;1) = \frac{\Gamma(c)\Gamma(c-a-b)}{\Gamma(c-a)\Gamma(c-b)}, \tag{6}$$

*where*

$$_2F_1(a,b;c;z) := \sum_{k=0}^{\infty} \frac{(a)_k(b)_k}{(c)_k}\frac{z^k}{k!}, \qquad _1F_1(a;b;z) := \sum_{k=0}^{\infty} \frac{(a)_k}{(b)_k}\frac{z^k}{k!}.$$

In future work, we will denote by $H_\tau(x)$ the *Hermite function (of degree $\tau$)*, which can be represented in terms of the confluent hypergeometric function $_1F_1$ as follows [7]:

$$H_\tau(x) = 2^\tau \frac{\Gamma(\frac{1}{2})}{\Gamma(\frac{1-\tau}{2})}\,_1F_1\left(-\frac{\tau}{2};\frac{1}{2};x^2\right) + 2^\tau x \frac{\Gamma(-\frac{1}{2})}{\Gamma(-\frac{\tau}{2})}\,_1F_1\left(\frac{1-\tau}{2};\frac{3}{2};x^2\right). \tag{7}$$

### 3. Integral Representation on $\mathbb{R}$

In the following result, we present a new definite integration formulae involving the Hermite functions.

**Lemma 4.** *For any $(z, \tau) \in \mathbb{C}^2$, with $\Re(z) > -1$, the following formulae hold:*

$$\int_0^\infty x^z H_\tau(x)e^{-x^2}dx = \frac{\sqrt{\pi}}{2^{z-\tau+1}}\frac{\Gamma(z+1)}{\Gamma(\frac{z-\tau}{2}+1)}, \tag{8}$$

$$\int_{-\infty}^\infty |x|^z H_\tau(|x|)e^{-x^2}dx = \frac{\sqrt{\pi}}{2^{z-\tau}}\frac{\Gamma(z+1)}{\Gamma(\frac{z-\tau}{2}+1)}. \tag{9}$$

**Proof.** Since the function $|x|^\nu H_\tau(|x|)e^{-x^2}$ is even, it is enough to prove (8).

Let us fix $\tau \in \mathbb{C}$, with $\Re(\tau) > -1$. For any $z \in \mathbb{C}$, with $-1 < \Re(z) < \Re(\tau)$, let us consider the following integral:

$$\Lambda(z) := \int_0^\infty x^z H_\tau(x)e^{-x^2}dx.$$

Using (7), the previous integral can be written as

$$\Lambda(z) = 2^\tau \frac{\Gamma(\frac{1}{2})}{\Gamma(\frac{1-\tau}{2})}\Pi(z) + 2^\tau \frac{\Gamma(-\frac{1}{2})}{\Gamma(-\frac{\tau}{2})}\Omega(z), \tag{10}$$

where

$$\Pi(z) := \int_0^\infty x^z {}_1F_1\left(-\frac{\tau}{2};\frac{1}{2};x^2\right)e^{-x^2}dx,$$

$$\Omega(z) := \int_0^\infty x^{z+1} {}_1F_1\left(\frac{1-\tau}{2};\frac{3}{2};x^2\right)e^{-x^2}dx.$$

By changing the variable of integration, by setting $t = x^2$, and using (5), with $s = 1$, $\alpha = (z+1)/2$, $a_1 = -\tau/2$, and $b_1 = 1/2$, we obtain

$$\Pi(z) = \frac{1}{2}\Gamma\left(\frac{z+1}{2}\right){}_2F_1\left(-\frac{\tau}{2},\frac{z+1}{2};\frac{1}{2};1\right).$$

Again, with (5), where $s = 1$, $\alpha = (z+2)/2$, $a_1 = (1-\tau)/2$, and $b_1 = 3/2$, we get

$$\Omega(z) = \frac{1}{2}\Gamma\left(\frac{z+2}{2}\right){}_2F_1\left(\frac{1-\tau}{2},\frac{z+2}{2};\frac{3}{2};1\right).$$

Since $\Re(z) < \Re(\tau)$, by using (6) $\Pi(z)$ and $\Omega(z)$ can be written as

$$\Pi(z) = \frac{\Gamma(\frac{z+1}{2})\Gamma(\frac{1}{2})\Gamma(\frac{\tau-z}{2})}{2\Gamma(\frac{1+\tau}{2})\Gamma(-\frac{z}{2})},$$

$$\Omega(z) = \frac{\Gamma(\frac{z+2}{2})\Gamma(\frac{3}{2})\Gamma(\frac{\tau-z}{2})}{2\Gamma(\frac{2+\tau}{2})\Gamma(\frac{1-z}{2})}.$$

Therefore, taking into account $\Gamma(\frac{1}{2})^2 = -\Gamma(-\frac{1}{2})\Gamma(\frac{3}{2}) = \pi$, the expression (10) can be rewritten as follows:

$$\Lambda(z) = \frac{2^{\tau-1}\pi\Gamma(\frac{\tau-z}{2})}{\Gamma(-\frac{z}{2})\Gamma(\frac{1-z}{2})}\Big(U(z,\tau) - U(z+1,\tau+1)\Big),$$

where

$$U(z,\tau) = \frac{\Gamma(\frac{z+1}{2})\Gamma(\frac{1-z}{2})}{\Gamma(\frac{1+\tau}{2})\Gamma(\frac{1-\tau}{2})}.$$

Using the duplication formula

$$\Gamma(u)\Gamma(1-u) = \frac{\pi}{\sin(\pi u)},$$

a straightforward calculation leads to

$$U(z,\tau) = \frac{\cos(\frac{\pi}{2}\tau)}{\cos(\frac{\pi}{2}z)}, \quad U(z+1,\tau+1) = \frac{\sin(\frac{\pi}{2}\tau)}{\sin(\frac{\pi}{2}z)}.$$

Then,

$$\Lambda(z) = -\frac{2^{\tau}\pi\Gamma(\frac{\tau-z}{2})}{\Gamma(-\frac{z}{2})\Gamma(\frac{1-z}{2})}\frac{\sin\left(\frac{\pi}{2}(\tau-z)\right)}{\sin(\pi z)},$$

so, by using the Gauss–Legendre multiplication formula,

$$\Gamma(u)\Gamma(u+\tfrac{1}{2}) = 2^{1-2u}\sqrt{\pi}\,\Gamma(2u),$$

and, again, with the duplication formula, we get

$$\Lambda(z) = \frac{\sqrt{\pi}}{2^{z-\tau+1}}\frac{\Gamma(z+1)}{\Gamma(1+\frac{z-\tau}{2})}.$$

For this proof, we assumed the conditions $-1 < \Re(z) < \Re(\tau)$, then the integral $\Lambda(z)$ converged exponentially to zero when $\tau \to \infty$. Hence, through analytic continuation, (10) is valid for each $(\tau, z) \in \mathbb{C}^2$, with $\Re(z) > -1$. $\square$

**Remark 1.** *Note that the above result also covers the $z = \tau$ case. In fact, if $\tau = 0, 1, \dots$ this identity represents the property of orthogonality for the monic Hermite polynomials.*

As a consequence, we have the following result:

**Corollary 1.** *For any $\tau \in \mathbb{C}$, with $\Re(\tau) > -1$, the following formulae hold:*

$$\int_0^\infty x^{2n+\tau}H_\tau(x)e^{-x^2}dx = \frac{\sqrt{\pi}}{2^{2n+1}}\frac{\Gamma(2n+\tau+1)}{\Gamma(n+1)}, \tag{11}$$

$$\int_{-\infty}^\infty x^{2n}|x|^\tau H_\tau(|x|)e^{-x^2}dx = \frac{\sqrt{\pi}}{2^{2n}}\frac{\Gamma(2n+\tau+1)}{\Gamma(n+1)}. \tag{12}$$

**Theorem 1.** *For any $\tau \in \mathbb{C}$, with $\Re(\tau) > -1$, the linear functional $\mathscr{G}_H(\tau)$ has the following integral representation:*

$$\langle \mathscr{G}_H(\tau), p \rangle = \frac{1}{\sqrt{\pi}\,\Gamma(\tau + 1)} \int_{-\infty}^{\infty} p(x)|x|^\tau H_\tau(|x|)e^{-x^2}dx, \quad p \in \mathbb{P}, \tag{13}$$

*where $H_\tau$ is the Hermite function (of degree $\tau$).*

**Proof.** Due to the Equation (1) and Corollary 1,

$$(\mathscr{G}_H(\tau))_{2n} = \frac{(\tau + 1)_{2n}}{n!2^{2n}} = \frac{\Gamma(2n + \tau + 1)}{2^{2n}\Gamma(n + 1)\Gamma(\tau + 1)}$$

$$= \frac{1}{\sqrt{\pi}\,\Gamma(\tau + 1)} \int_{-\infty}^{\infty} x^{2n}|x|^\tau H_\tau(|x|)e^{-x^2}dx,$$

$$(\mathscr{G}_H(\tau))_{2n+1} = 0 = \frac{1}{\sqrt{\pi}\,\Gamma(\tau + 1)} \int_{-\infty}^{\infty} x^{2n+1}|x|^\tau H_\tau(|x|)e^{-x^2}dx.$$

Therefore, one has

$$(\mathscr{G}_H(\tau))_n = \frac{1}{\sqrt{\pi}\,\Gamma(\tau + 1)} \int_{-\infty}^{\infty} x^n|x|^\tau H_\tau(|x|)e^{-x^2}dx, \qquad n = 0, 1, \ldots$$

Consequently, for any polynomial $p \in \mathbb{P}$,

$$\langle \mathscr{G}_H(\tau), p \rangle = \frac{1}{\sqrt{\pi}\,\Gamma(\tau + 1)} \int_{-\infty}^{\infty} p(x)|x|^\tau H_\tau(|x|)e^{-x^2}dx.$$

$\square$

Observe that if we set $n = 0$ in (11) we get a new integral representation for the Euler Gamma function. In fact, for any $\tau \in \mathbb{C}$, with $\Re(\tau) > -1$,

$$\Gamma(\tau + 1) = \frac{2}{\sqrt{\pi}} \int_0^{\infty} x^\tau H_\tau(x)e^{-x^2}dx, \tag{14}$$

$$\Gamma(\tau + 1) = \frac{1}{\sqrt{\pi}} \int_{-\infty}^{\infty} |x|^\tau H_\tau(|x|)e^{-x^2}dx. \tag{15}$$

## 4. Integral Representation on the Complex Plane

**Theorem 2.** *For any $\tau \in \mathbb{C}$, the following identities hold:*

*(i)*

$$\int_{\mathbf{C_1}} \zeta^{2n+1}|\zeta|^\tau H_\tau(|\zeta|)e^{-\zeta^2}d\zeta = 0, \quad n = 0, 1, \ldots$$

*(ii)* *For any $n \in \mathbb{N}$, so that $\tau + 2n + 1$ is not a negative integer, we have*

$$\int_{\mathbf{C_1}} \zeta^{2n}|\zeta|^\tau H_\tau(|\zeta|)e^{-\zeta^2}d\zeta = -\frac{\sqrt{\pi}}{2^{2n}}\frac{\Gamma(2n + \tau + 1)}{\Gamma(n + 1)}, \quad n = 0, 1, \ldots$$

*where $\mathbf{C_1}$ is the following contour in the complex plane (See Figure 1).*

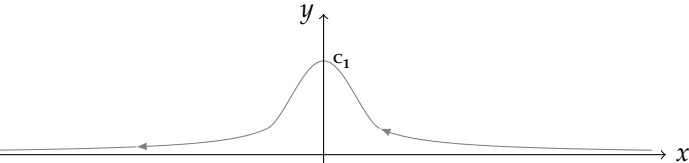

**Figure 1.** Path $\mathbf{C_1}$.

**Proof.** We deform $\mathbf{C_1}$ into a contour $\tilde{\mathbf{C}}_1$ consisting of two straight lines and a circle (see Figure 2).

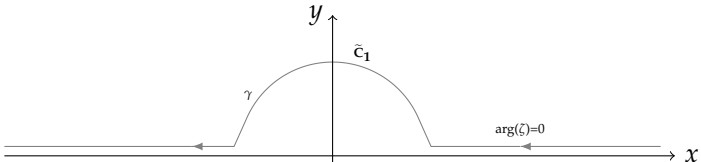

**Figure 2.** Path $\tilde{\mathbf{C}}_1$.

where $\gamma := \{\zeta \in \mathbb{C} : \Im(\zeta) > 0, |\zeta| = \epsilon\}$, being $\epsilon > 0$.

Now, for each integer $n \geq 0$ and $\tau \in \mathbb{C}$, we define

$$
\begin{aligned}
I_n(\tau) := \quad & \int_{\tilde{\mathbf{C}}_1} \zeta^n |\zeta|^\tau H_\tau(|\zeta|) e^{-\zeta^2} d\zeta = \int_\infty^\epsilon \zeta^n |\zeta|^\tau H_\tau(|\zeta|) e^{-\zeta^2} d\zeta \\
& + \int_\gamma \zeta^n |\zeta|^\tau H_\tau(|\zeta|) e^{-\zeta^2} d\zeta + \int_{-\epsilon}^{-\infty} \zeta^n |\zeta|^\tau H_\tau(|\zeta|) e^{-\zeta^2} d\zeta.
\end{aligned}
$$

So, if $\Re(\tau) > -n-1$, after a direct computation, we get

$$
\begin{aligned}
\lim_{\epsilon \to 0} \int_\infty^\epsilon \zeta^n |\zeta|^\tau H_\tau(|\zeta|) e^{-\zeta^2} d\zeta &= -\int_0^\infty x^{n+\tau} H_\tau(x) e^{-x^2} dx, \\
\lim_{\epsilon \to 0} \int_{-\epsilon}^{-\infty} \zeta^n |\zeta|^\tau H_\tau(|\zeta|) e^{-\zeta^2} d\zeta &= -(-1)^n \int_0^\infty x^{n+\tau} H_\tau(x) e^{-x^2} dx.
\end{aligned}
$$

For the middle integral, we obtain

$$
\begin{aligned}
\left| \int_\gamma \zeta^n |\zeta|^\tau H_\tau(|z|) e^{-\zeta^2} d\zeta \right| &= \left| \int_0^\pi \epsilon^n e^{in\theta} \epsilon^\tau H_\tau(\epsilon) e^{-\epsilon^2 e^{2i\theta}} \epsilon i e^{i\theta} d\theta \right| \\
&\leq \epsilon^{n+\Re(\tau)+1} \int_0^\pi |H_\tau(\epsilon)| e^{-\epsilon^2 \cos(2\theta)} d\theta,
\end{aligned}
$$

knowing that $H_\tau(0) = 2^\tau \sqrt{\pi}/\Gamma(\frac{1-\tau}{2})$, it is straightforward to see that

$$
\lim_{\epsilon \to 0} \int_\gamma \zeta^n |\zeta|^\tau H_\tau(|\zeta|) e^{-\zeta^2} d\zeta = 0.
$$

Therefore, for each $n \geq 0$ and $\tau \in \mathbb{C}$, such that $\Re(\tau) > -n-1$, we have

$$
I_n(\tau) = -\left((-1)^n + 1\right) \int_0^\infty x^{n+\tau} H_\tau(x) e^{-x^2} dx.
$$

Then, $I_{2n+1}(\tau) = 0$ for all $n \geq 0$. Notice that for the proof of i), we assumed $\Re(\tau) > -n-1$, but the integral converges exponentially when $\tau \to \infty$, and therefore it exists for all $\tau$. Hence, (i) holds through analytic continuation for any $\tau \in \mathbb{C}$.

On the other hand, using (11), it follows that

$$
I_{2n}(\tau) = -\frac{\sqrt{\pi}}{2^{2n}} \frac{\Gamma(2n+\tau+1)}{\Gamma(n+1)}.
$$

Hence, (ii) holds, for the same reason already quoted and by analytic continuation of $\tau \in \mathbb{C}$, except when $2n + \tau + 1$ is a negative integer, where the function $\Gamma$ is undefined. $\square$

As a consequence, we have the following result.

**Theorem 3.** *For any $\tau \in \mathbb{C}$, with $-\tau \notin \mathbb{N}$, the linear functional $\mathscr{G}_H(\tau)$ has the following integral representation:*

$$\langle \mathscr{G}_H(\tau), p \rangle = -\frac{1}{\sqrt{\pi}\Gamma(\tau+1)} \int_{\mathbf{C_1}} p(x)|x|^\tau H_\tau(|x|)e^{-x^2}dx, \quad p \in \mathbb{P}, \tag{16}$$

*where $H_\tau$ is the Hermite function (of degree $\tau$).*

Using an analog idea allows us to formulate another integral representation for the gamma function in the complex plane by using a different contour.

**Theorem 4.** *For any $\tau \in \mathbb{C}$, with $-\tau \notin \mathbb{N}$, the Euler's Gamma function satisfies the following integral representation:*

$$\Gamma(\tau+1) = \frac{2}{\sqrt{\pi}(e^{2\pi i\tau}-1)} \int_{\mathbf{C}} \zeta^\tau H_\tau(\zeta)e^{-\zeta^2}d\zeta, \tag{17}$$

*where C is the following contour in the complex plane (See Figure 3).*

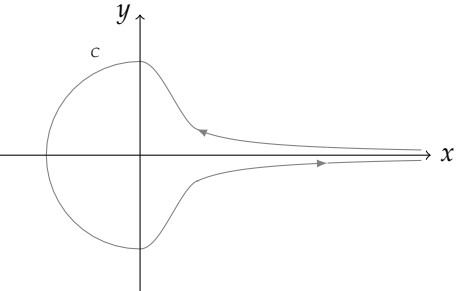

**Figure 3.** Path of integration $C$.

**Proof.** We deform $C$ into a contour $\tilde{C}$ consisting of two straight lines and a circle (See Figure 4):

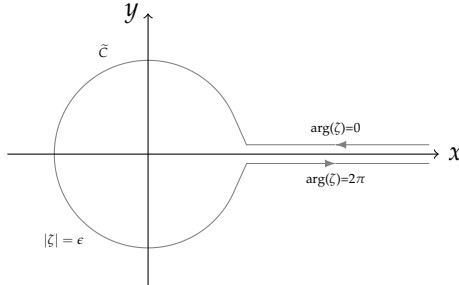

**Figure 4.** Path of integration $\widetilde{C}$.

We let

$$J(\tau) = \int_{\tilde{\mathbf{C}}} \zeta^\tau H_\tau(\zeta)e^{-\zeta^2}d\zeta$$

Then

$$J(\tau) = \int_\infty^\epsilon \zeta^\tau H_\tau(\zeta)e^{-\zeta^2}d\zeta + \int_{|\zeta|=\epsilon} \zeta^\tau H_\tau(\zeta)e^{-\zeta^2}d\zeta + \int_\epsilon^\infty \zeta^\tau H_\tau(\zeta)e^{-\zeta^2}d\zeta,$$

and if $\Re(\tau) > -1$ in a direct way, we obtain

$$\lim_{\epsilon \to 0} \int_{\infty}^{\epsilon} \zeta^{\tau} H_{\tau}(\zeta) e^{-\zeta^2} d\zeta = -\frac{\sqrt{\pi}}{2} \Gamma(\tau + 1),$$

$$\lim_{\epsilon \to 0} \int_{\epsilon}^{\infty} \zeta^{\tau} H_{\tau}(\zeta) e^{-\zeta^2} d\zeta = e^{2\pi i \tau} \frac{\sqrt{\pi}}{2} \Gamma(\tau + 1).$$

For the middle integral, we obtain

$$\left| \int_{|\zeta| = \epsilon} \zeta^{\tau} H_{\tau}(\zeta) e^{-\zeta^2} d\zeta \right| = \left| \int_{0}^{2\pi} (\epsilon e^{i\theta})^{\tau} H_{\tau}(\epsilon e^{i\theta}) e^{-\epsilon^2 e^{2i\theta}} \epsilon i e^{i\theta} d\theta \right|$$

$$\leq \epsilon^{\Re(\tau) + 1} \int_{0}^{2\pi} |H_{\tau}(\epsilon e^{i\theta})| e^{-\epsilon^2 \cos(2\theta) - \theta\left(\Im(\tau) + 1\right)} d\theta,$$

thus,

$$\lim_{\epsilon \to 0} \int_{|\zeta| = \epsilon} \zeta^{\tau} H_{\tau}(\zeta) e^{-\zeta^2} d\zeta = 0.$$

Finally,

$$J(\tau) = (e^{2\pi i \tau} - 1) \frac{\sqrt{\pi}}{2} \Gamma(\tau + 1),$$

hence, the result holds. In the proof, we have assumed that $\Re(\tau) > -1$, but the integral (17) converges exponentially at infinity, and therefore it exists for all $\tau$. In fact, through analytic continuation, the result is valid for every complex $\tau$, except for the negative integers, where the denominator vanishes. $\square$

In addition, from the last representation, we obtain the following:

$$\Gamma(\tau + 1) = \frac{1}{i\sqrt{\pi} \sin(\pi\tau)} \int_{\mathbf{C}} (-\zeta)^{\tau} H_{\tau}(\zeta) e^{-\zeta^2} d\zeta.$$

In the last result, we show a representation for the reciprocal of $\Gamma(\tau + 1)$.

**Theorem 5.**

$$\frac{1}{\Gamma(\tau + 1)} = -i\pi^{-\frac{3}{2}} \int_{\mathbf{C}} (-\zeta)^{-1-\tau} H_{-1-\tau}(\zeta) e^{-\zeta^2} d\zeta.$$

*This representation is valid for all $\tau$ and C is the same contour as in the previous theorem.*

**Proof.** Based on the last representation, one has

$$\Gamma(-\tau) = \frac{1}{i\sqrt{\pi} \sin(\pi\tau)} \int_{\mathbf{C}} (-\zeta)^{-1-\tau} H_{-1-\tau}(\zeta) e^{-\zeta^2} d\zeta$$

$$= \frac{\Gamma(\tau + 1)\Gamma(-\tau)}{i\pi^{\frac{3}{2}}} \int_{\mathbf{C}} (-\zeta)^{-1-\tau} H_{-1-\tau}(\zeta) e^{-\zeta^2} d\zeta.$$

This leads to the desired result. $\square$

## 5. Conclusions

We have obtained integral representations of a generalized linear Hermite functional, which is among the natural extensions of the linear Hermite functional, using the fact this linear functional is symmetric, i.e., the odd moments associated with this functional are zero, and also the fact that some hypergeometric representations associated with the Hermite polynomials are known. Observe that this can also be implemented for other symmetric classical orthogonal polynomials. Moreover, we have obtained an integral representation for the generalized linear Hermite functional in the complex plane, and from

this integral representation, we are able to obtain a novel integral representation for the Euler Gamma function.

Of course, this method can be applied not only to other (symmetric) classical orthogonal polynomials but to any other symmetric orthogonal polynomial sequence for which a hypergeometric representation is known. This is something we should do in order to obtain novel integral representations for other Special functions; for example we could consider some other generalization for the Hermite linear functional, as well as some Laguerre–Hahn or semi-classical, orthogonal polynomials (see, e.g., [8,9] and the references therein).

**Funding:** This work was funded by Universidad Loyola Andalucía.

**Conflicts of Interest:** The author declares no conflict of interest.

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
