# Peer review of "Integral Representations of a Generalized Linear Hermite Functional"

_mathematics, doi:10.3390/math11143227_

Round 1

Reviewer 1 Report

The author obtains new integral representations for a generalized  Hermite linear functional in the real line and the complex plane.  The generalized linear functional is constructed by term by term  multiplication  of the linear functional associated with the sequence of moments of the classical Hermite polynomials  and a sequence that depends on one parameter.

Some new integral representations for the Gamma function are also obtained. 

The paper will be of interest to researchers in the areas of orthogonal polynomials, special functions and complex analysis. 

The results are relevant and the proofs are sufficiently detailed.  In the  proofs  the author uses  some properties of  hypergeometric functions and methods  from complex analysis to deal with the integral representations. 

The author does not mention  if there are other types of linear functionals for which the methods used in this paper could be applied to obtain integral representations.

Author Response

Thank you very much for spending your time doing the review of my work. This is very much appreciated. I added a short conclusion which, I believe, reply to your last comment.

Reviewer 2 Report

This paper is interesting.

Author Response

Thank you very much for spending your time reviewing my work. Regarding your report:

1) As far as I know, and I have done an extensive search no other integral representation for such generalization is known in the literature, but if for some reason my effort was not good enough and you know some, please share it with me, and I will add them in this work. 

2) There are no differences really between the integral representation I am obtaining and the rest as far as I am aware. It is just a symmetric case, and we use the information about the Hermite polynomials, which allows me to do the calculations. 

3) The relevance of the Euler Gamma function is so huge in the literature about Special functions that I don't think this is needed.

4) Sometimes, the novel results don't need to have complex backgrounds, but to see the idea, one must apply, which opens the way for further ideas. However, from my perspective, the knowledge of the Hermite polynomials, their hypergeometric series representation, and the properties of the even/odd functions are not simple at all. 

5) It would have been great if you could have told me what they are and where they are so that I could correct them.

Thank you again ;)

Reviewer 3 Report

See pdf file

Author Response

Thank you very much for spending your time reviewing my work. Regarding your report: 

1) About the ordering, usually, the reader has some mathematical background about orthogonal poñynomials, but it is presented in Section 2, where the set of polynomials in one variable $\mathbb P$ is defined, which answers your second comment.

3)  Amended.

4) Even though I don´t think the current work needs a Section entitled "conclusion" I added such section where some extra details which other referees commented about my work were added.